# Severity- and Time-Dependent Activation of Microglia in Spinal Cord Injury

**DOI:** 10.3390/ijms24098294

**Published:** 2023-05-05

**Authors:** Elvira Ruslanovna Akhmetzyanova, Margarita Nikolaevna Zhuravleva, Anna Viktorovna Timofeeva, Leisan Gazinurovna Tazetdinova, Ekaterina Evgenevna Garanina, Albert Anatolevich Rizvanov, Yana Olegovna Mukhamedshina

**Affiliations:** 1OpenLab Gene and Cell Technology, Institute of Fundamental Medicine and Biology, Kazan Federal University, 420008 Kazan, Russia; elyaelya18@gmail.com (E.R.A.);; 2Department of Morphology and General Pathology, Institute of Fundamental Medicine and Biology, Kazan Federal University, 420008 Kazan, Russia; 3Department of Histology, Cytology, and Embryology, Kazan State Medical University, 420012 Kazan, Russia

**Keywords:** spinal cord injury, microglia, polarization, proliferative and phagocytic activity, severity of injury, rat

## Abstract

A spinal cord injury (SCI) initiates a number of cascades of biochemical reactions and intercellular interactions, the outcome of which determines the regenerative potential of the nervous tissue and opens up capacities for preserving its functions. The key elements of the above-mentioned processes are microglia. Many assumptions have been put forward, and the first evidence has been obtained, suggesting that, depending on the severity of SCI and the post-traumatic period, microglia behave differently. In this regard, we conducted a study to assess the microglia behavior in the model of mild, moderate and severe SCI in vitro for various post-traumatic periods. We reported for the first time that microglia make a significant contribution to both anti- and pro-inflammatory patterns for a prolonged period after severe SCI (60 dpi), while reduced severities of SCI do not lead to prolonged activation of microglia. The study also revealed the following trend: the greater the severity of the SCI, the lower the proliferative and phagocytic activity of microglia, which is true for all post-traumatic periods of SCI.

## 1. Introduction

The primary post-traumatic responses that occur after spinal cord injury (SCI) are partly protective mechanisms of the body aimed at minimizing the consequences of injury. However, the severity and extent of the above-mentioned reactions can play a negative role, reducing the success in restoring the functions of the nervous tissue. It is considered that the modulation of microglia toward a neuroprotective or neurotoxic phenotype is one of the key processes that determine the outcome of post-traumatic reactions in the spinal cord [1,2].

The first mixed glial cultures were obtained in 1980 [3]. Since then, glial cells, in particular, microglia, have become the object of the close attention of researchers. To date, there are quite a few works describing changes in the phenotype of microglia [4,5]. Despite the incomplete objectivity, which does not reflect a wide range of phenotypic diversity, two main pathways of microglia polarization are distinguished: classically activated (M1) and alternatively activated (M2) [6,7]. Microglia-produced pro-inflammatory cytokines, such as tumor necrosis factor-α (TNF-α), interleukin-1 beta (IL-1β), interleukin-6 (IL-6), glutamate, superoxide, nitric oxide (NO), reactive oxygen species (ROS), chemokines and proteases form a neurotoxic environment [8,9]. At the same time, anti-inflammatory cytokines IL-4, IL-10, IL-13, transforming growth factor β (TGF-β) and neurotrophic factors such as ciliary neurotrophic factor (CNTF), insulin-like growth factor (IGF), epidermal growth factor (EGF) and nerve growth factor (NGF) produced by M2 microglia inhibit pro-inflammatory reactions that takes place after SCI [8,9].

The previous studies show the different effects that microglia exert in SCI. For example, remote activation of microglia was shown to aggravate allodynia in the acute period after SCI (7 dpi) in the model of mild and moderate rat SCI [10]. Ref. [11] demonstrated a long-term reduction in the pro-inflammatory response and preservation of a pro-regenerative milieu from 7 to 63 dpi of mice SCI after early administration of the inhibitor of microglia activation. These results confirm the earlier work indicating the active involvement of microglia in the maintenance of chronic pain after SCI on the rat model [12]. At the same time, it was established that an increase in the number of actively phagocytic microglia in the area of acute rat SCI may contribute to the improved nervous tissue integrity [13].

These results supported the hypothesis that an early resolution of the initial traumatic events can improve the long-term structural outcome. Other studies also show a decrease in neuronal survival and increase in axonal dieback caused by the depletion of microglia up to 7 days after mice SCI [14]. The above controversial results may suggest that the activation of microglia is not a definitive phenomenon and that there are several different states of “activation” in which microglia can selectively exert a neurotoxic or neuroprotective effect. This phenomenon was demonstrated in various studies, which confirm that the behavior of microglia is dependent on activation factors regulated by their microenvironment [15,16]. We assume that these activation factors may differ based on the post-traumatic period and degree of spinal cord injury. In this regard, we conducted a study to evaluate the behavior of microglia in the model of mild, moderate and severe severity SCI in vitro for various post-traumatic periods, as well as under the conditions of adding spinal cord extract (SCE) from intact spinal cord (ISC).

## 2. Results

### 2.1. Morphology and Phenotype of Cultivated Microglia

In the 24 h after isolation and cultivation under standard conditions, microglia showed a predominantly amoeboid morphology. On days 4–5, among single glial and fibroblast-like cells, we observed mainly microglia with elongated soma and 1–2 short processes (Figure 1A), which corresponds to the morphology of rod microglia described earlier [17]. Isolated microglia expressed Iba1, and among them, Iba1^+^/TNF-α^+^ and Iba1^+^/TGF-β^+^ cells were also found (Figure 1B–D).

Next, we modeled SCI in vitro by adding the appropriate SCE (Figure 2).

It was shown that more than 95% of microglia expressed CD86 in standard culture conditions (Figure 3A). In the model of SCI in vitro, the level of CD86 expression by these cells remains consistently high. However, significant differences were found when adding SCE samples obtained at 7 and 60 dpi, where the percentage of CD86^+^ cells was higher (*p* < 0.05) in SCI 1.5 and SCI 2.5 groups compared to the SCI 4 group.

Under standard culture conditions, slightly more than half of the obtained microglia expressed CD45 (55 ± 6%) (Figure 3B). When ISC samples were added to the microglia culture, a decrease (*p* < 0.05) in the expression of this marker was found. However, in the case of modeling the acute period of SCI in vitro (3 and 7 dpi), the level of CD45 expression approached the corresponding value in the Medium group but was significantly higher (*p* < 0.05) in the SCI 1.5 and SCI 4 groups compared to the ISC group. The maximum value of CD45 expression (61 ± 7%) was observed in modeling the subacute period of SCI in vitro (14 dpi) in the SCI 1.5 group. According to the increasing severity of SCI during this period, there was a tendency to decrease in CD45 expression with a minimum value in the SCI 4 group, close to that in the ISC group (SCI 4 43 ± 3% vs. ISC 40 ± 2%). In the model of the chronic period of SCI in vitro (60 dpi), there were no significant changes in the expression of CD45 in the study groups when compared to the Medium group. However, this score was higher (*p* < 0.05) in the SCI 2.5 group compared to the SCI 1.5, SCI 4 and ISC groups.

Among the obtained microglia, cultivating under standard conditions, the number of CD206+ cells turned out to be extremely small (4 ± 2%) (Figure 3C). However, when ISC samples were added, the CD206 expression significantly increased (6.5 times) (*p* < 0.05) compared to the Medium group. When SCE samples obtained at 3 dpi were added, the CD206 expression in the SCI 1.5 and SCI 2.5 groups was similar to the ISC group and significantly higher (*p* < 0.05) than the Medium group. In the conditions of modeling a 7 dpi SCI in vitro, the CD206 expression in all groups with the addition of SCE obtained from the injured spinal cord was higher (*p* < 0.05) compared to the ISC and Medium groups. It should be noted that the highest level of CD206 expression in the model of the acute period of SCI in vitro (3 and 7 dpi) was found in the SCI 4 group. When modeling the subacute (14 dpi) and chronic (60 dpi) periods of SCI in vitro, the percentage of CD206^+^ cells in the SCI 1.5 group sharply decreased when compared with that in the acute period (3 and 7 dpi), while there were no substantial changes in the CD206^+^ cell count in the SCI 2.5 and SCI 4 groups.

### 2.2. Effect of SCE on Microglia Gene Expression

Under the conditions of the SCE samples in the SCI 1.5, SCI 2.5 and SCI 4 groups obtained from 3, 7, 14 and 60 dpi addition, as well as an SCE sample of ISC to microglia on days 4–5 of isolation and cultivation with further cultivation with SCE for 72 h, the changes in the expression of M1 and M2 microglia-specific gene mRNAs varied depending on the severity and the period after injury.

There was a significantly (*p* < 0.05) high expression of microglia pan marker *Iba1* (Figure 4A) in the model of acute period SCI in vitro (3 dpi) in the SCI 4 group when compared to SCI 1.5 and SCI 2.5 and Medium groups. In the course of further post-traumatic processes in vitro (7 dpi), the situation changed and a sharp increase in the expression of *Iba1* mRNA in the SCI 1.5 group with significant differences (*p* < 0.05) with other experimental groups in each period was observed. A similar pattern in the acute period of SCI in vitro (3 and 7 dpi) was found in terms of the level of *TGF-β* mRNA expression (Figure 4B). In the model of the subacute (14 dpi) period of SCI in vitro, there were no significant differences between the experimental groups in terms of the relative expression of *Iba1* mRNA. However, by 60 dpi, there was an increase in the expression of *Iba1* mRNA in the SCI 2.5 group compared to other experimental groups in this post-traumatic period.

The relative expression of *Il-1β* mRNA (Figure 4C) significantly decreased when adding SCE of the intact and injured spinal cord for all studied periods, with the exception of the SCI 4 group, where this rate was higher in the acute period of SCI in vitro (3 dpi) (*p* < 0.05) compared to other experimental groups, including the Medium group. The relative expression of *Il-6* mRNA (Figure 4D) in the model of acute (3 dpi) and chronic (60 dpi) periods of SCI in vitro significantly increased and was maximal (*p* < 0.05) in the SCI 4 group compared to other experimental groups. However, in the subacute (14 dpi) period of SCI in vitro, this indicator, on the contrary, significantly decreased and was minimal (*p* < 0.05) in the SCI 4 group compared to the SCI 1.5 and SCI 2.5 groups.

We found that the relative expression of *TNF-α* mRNA (Figure 4E) in the model of the acute period of SCI in vitro (3 dpi) was significantly higher in the SCI 2.5 group compared to other experimental and control groups in different post-traumatic periods. During the acute (7 dpi) and subacute (14 dpi) periods of SCI in vitro, the relative level of *TNF-α* mRNA expression decreased in the SCI 2.5 group but increased in the SCI 1.5 and SCI 4 groups, respectively, with differences (*p* < 0.05) in the Medium and ISC groups.

The relative expression of *CD209* mRNA (Figure 4F) in microglia in an in vitro model of the acute period of SCI (3 dpi) was higher (*p* < 0.05) in the SCI 4 group compared to the SCI 1.5 and ISC groups and showed similar indicators by 7, 14 and 60 dpi in all experimental groups. The analysis of *CCL22* mRNA expression (Figure 3G) showed the highest values in the SCI 4 group at 3, 14 and 60 dpi, while the corresponding parameter in the model of SCI (7 dpi), on the contrary, was significantly reduced (*p* < 0.05) in the SCI 4 group compared to other experimental groups in different post-traumatic periods.

### 2.3. Effect of SCE on Cytokine Profile of Microglia

Adding SCE of the intact spinal cord (ISC group) to the microglia culture leads to significant changes in the expression of the following cytokines: IL-6, MIP-1α, VEGF and MCP-1 by these cells (Figure 5; Appendix A).

In the model of different periods of SCI in vitro, changes in the cytokine profile expression of microglia were also observed. Therefore, the level of the pro-inflammatory cytokine IL-1α was significantly increased in the SCI 1.5 and SCI 4 groups at 7 and 60 dpi compared to the ISC and Medium groups (Figure 6A). At the same time, the expression of another pro-inflammatory cytokine, IL-2, greatly increased (*p* < 0.05) in the model of the acute (3 and 7 dpi) and subacute (14 dpi) period of severe SCI (SCI 4 group) compared to other groups at the appropriate period of SCI in vitro. However, in the chronic period (60 dpi), this indicator significantly increased in the model of mild SCI (SCI 1.5 group) compared to the SCI 2.5 and SCI 4 groups.

The level of the pro-inflammatory chemokine GRO/KC significantly (*p* < 0.05) increased in the acute period model (3 dpi) in the SCI 2.5 group compared to the other experimental and control groups (Figure 6B). However, this rate in the delayed period (7 and 14 dpi) was increased in the SCI 4 groups, remaining so at 60 dpi and increasing in the same period in the SCI 1.5 group. The activator of pro-inflammatory processes, RANTES chemokine, was found to be overexpressed (*p* < 0.05) in the acute period (3 dpi) in the SCI 2.5 group and in the delayed post-traumatic periods (7, 14 and 60 dpi) in the SCI 1.5 and SCI 4 groups compared to the ISC and Medium groups.

In the model of 7, 14 and 60 dpi SCI in vitro, the level of expression of microglia proliferation and activation factor G-CSF was higher in the SCI 1.5 and SCI 4 groups with significant differences from the ISC and Medium groups and, possibly, from the SCI 2.5 group, for which a similar indicator was not detectable with the kit. The level of expression of MIP-1α and IL-6, which are involved in the pro-inflammatory response of microglia, was consistently high in the SCI 4 group in all investigated post-traumatic periods, while in the SCI 1.5 group, the above indicator had the same high values at 3 and 60 dpi (Figure 6C,D). MIP-3α, which performs functions similar to MIP-1α, tended to increase mainly in the chronic period (60 dpi) of mild and severe SCI but showed low expression by microglia in the SCI 4 group in the acute period (3 dpi) (Figure 6E). The expression level of the soluble marker of the neurotoxic microglia TNF-α was elevated in the acute period (7 dpi) in the SCI 1.5 group and also significantly increased (*p* < 0.05) in the chronic period (60 dpi) of mild and severe SCI compared to the ISC and Medium groups. Similar expression rates were observed for the pro-inflammatory cytokine MCP-1 and vascular endothelial growth factor (VEGF), which increased (*p* < 0.05) in the SCI 4 group at 7, 14 and 60 dpi compared to the ISC and Medium groups (Figure 6F). However, in contrast to VEGF, MCP-1 also exhibited an increased (*p* < 0.05) level of expression in the SCI 2.5 group on 3 and 14 dpi.

Interestingly, the expression of most of the studied anti-inflammatory cytokines (IL-4, IL-13, IL-17) did not change significantly depending on the severity and period of SCI in vitro. However, the level of IL-10 expression increased sharply (*p* < 0.05) in the model of acute (7 dpi), subacute (14 dpi), and chronic (60 dpi) periods of severe SCI in vitro.

### 2.4. Effect of SCE on Microglia Proliferation

The proliferation of microglia was increased in all the experimental groups (SCI 1.5, SCI 2.5 and SCI 4 groups) compared to the ISC group in the model of the acute period of SCI in vitro (3 dpi) (Figure 7A). Moreover, the milder the severity of SCI, the higher the proliferation of microglia. Under the conditions of 7 dpi SCI in vitro, the studied parameter did not show significant differences between the experimental groups, while the proliferation of microglia in groups with the addition of various SCEs was lower (*p* < 0.05) compared to the Medium group, starting from 80 h of cultivation (Figure 7B).

In the model of the subacute period of SCI in vitro (14 dpi), the proliferation of microglia significantly decreased in the SCI 2.5, SCI 4 and ISC groups, while the addition of SCE in SCI 1.5 group did not lead to changes in the above-mentioned rate, which was the same as the standard conditions of cultivation (Medium group) (Figure 7C). Under conditions of chronic (60 dpi) SCI in vitro, the addition of SCE in the SCI 1.5, SCI 2.5 and SCI 4 groups no longer resulted in significant changes in the proliferation of microglia (Figure 7D).

### 2.5. Effect of SCE on Microglia Phagocytic Activity

After a 1-h incubation with SCE, microglia showed the highest level of phagocytic activity in the model of various post-traumatic periods of mild SCI in vitro (SCI 1.5 group) compared to other experimental groups at the same time points (Figure 8A). It was found that the more severe the degree of damage to the spinal cord, the lower the phagocytic activity of the microglia.

After 24 h of cultivation with SCE, when modeling acute (3 dpi) and subacute (14 dpi) periods of SCI in vitro, the phagocytic activity of microglia increased (*p* < 0.05) in the SCI 1.5 group compared to the SCI 2.5, SCI 4 and ISC groups but did not show significant differences when compared with standard culture conditions (Medium group) (Figure 8B). In the model of the acute period (7 dpi) of SCI in vitro, the highest value of the phagocytic activity of microglia was also found in the SCI 1.5 group, with differences from the SCI 4 group being determined. Under the conditions of modeling the chronic period of SCI in vitro (60 dpi), the phagocytic activity of microglia was higher (*p* < 0.05) in the SCI 1.5 group compared to the SCI 2.5 and ISC groups.

After 72 h of cultivation with SCE, the index of the phagocytic activity of the microglia was close to that obtained after 24 h of cultivation. Thus, when modeling various post-traumatic periods of SCI in vitro, the studied rate in the SCI 1.5 group was the highest (Figure 8C). At the same time, there was a decrease in the phagocytic activity of microglia with an increase in the severity of SCI, approaching the same indicator in the ISC group.

## 3. Discussion

A simplified version of the classification that distinguishes populations of microglia with a neurotoxic (M1) or protective (M2) phenotype indicates differences in the secretome and transcriptional profile of these cells. Of course, this approach does not reflect a wide range of phenotypic diversity but still facilitates our understanding of the reactive states of these cells and their plasticity in response to various stimulating signals in the microenvironment. It is still unknown how temporary (post-traumatic period) and structural–functional (depending on the degree of damage) changes in the microenvironment of microglia affect the shift of the polarization of these cells in one direction or another. Assuming that the ratio of phenotypically different populations of microglia may change at different periods after SCI, in our work, we investigated the behavior of the above-mentioned cells in the model of mild (SCI 1.5), moderate (SCI 2.5) and severe (SCI 4) severity in acute (3 and 7 dpi), subacute (14 dpi) and chronic (60 dpi) periods of SCI in vitro.

For modeling SCI in vitro, microglia from the cerebral cortex of newborn rat pups were obtained. More than 95% of microglia express CD86 under standard culture conditions, as shown by flow cytometry. To determine the M1/M2 ratio of microglia under conditions of modeling SCI in vitro, the percentage of CD86^+^, CD45^+^ and CD206^+^ cells was analyzed. We found that the number of CD86^+^ cells remained consistently high when modeling SCI of varying severity in vitro, but the lowest values were found in the group with severe SCI and ISC. Recently, [18] showed that CD86 is constitutively expressed by all microglia, and the intermediate M_1/2_ phenotype of microglia is characterized by the simultaneous expression of CD86 and CD206. It should be noted that the percentage of M2 microglia (CD206^+^), on the contrary, was the highest in the group with severe SCI, tending to decrease with the mitigation of the degree of injury. All of the above may indicate the suppression of the neurotoxic phenotype of microglia in the case of severe SCI during a long post-traumatic period (up to 60 dpi). With regard to CD45 expression, it is a leukocyte marker that determines mature macrophages or microglia [19]. It is known that an increase in CD45 may indicate the activation of microglia [20], which is consistent with our results obtained from in vitro modeling of SCI under conditions of adding injured SCE but not intact ones.

Comparing the data obtained during flow cytometry and RT-PCR, we can talk about the fact that the greatest changes in the phenotype of microglia, depending on the degree of damage, are in the model of SCI in vitro at 3, 14 and 60 dpi. Thus, in the acute period of severe SCI in vitro (3 dpi), a shift in the polarization of microglia towards a neuroprotective phenotype was noted, given the increase in the number of CD206^+^ cells against the background of high expression of CD86 (surface markers of M2 and M1 microglia phenotypes, respectively). However, despite a significant increase in mRNA expression of *TGF-β*, *CCL22* and *CD209* (soluble and surface markers of M2 microglia, respectively), we also found high expression levels of the mRNA of *Il-1β* and *IL-6* (soluble markers of M1 microglia), which, in our opinion, indicate active attempts to polarize microglia in one direction or another and the presence of an intermediate M_1/2_ phenotype in the acute period of SCI (3 dpi), consistent with the previously obtained results in human SCI in vivo [21]. When modeling the subacute (14 dpi) and chronic (60 dpi) periods of severe SCI in vitro, we also found a shift in the polarization of microglia towards an intermediate phenotype (CD206^+^/CD86^+^ cells) but noted, at the same time, a high level of mRNA expression of *CCL22* and *Il-1β* in both time points with opposite changes in mRNA expression of *IL-6* and *TNF-α* at 14 dpi. Thus, it can be noted that in the delayed periods after SCI, microglia make a significant contribution to both anti- and pro-inflammatory patterns in the case of severe damage, while milder SCI variants do not lead to prolonged activation of the studied cells.

To validate the above results, a study of changes in the secretory profile of microglia under conditions of SCI in vitro modeling was made. Thus, high rates of expression of both pro-inflammatory (IL-1β and IL-6) and anti-inflammatory cytokines (IL-10, IL-4 and IL-17) by microglia were confirmed in the model of the chronic period (60 dpi) of severe SCI in vitro. The above results can be considered as further evidence for the persistence of microglia activation in the chronic period of severe SCI. Previously [22], in the rat model, showed that as the severity of SCI increases, there is an increase in the expression of pro-inflammatory cytokines (IL-1β, IL-6 and TNF-α) in the first 3 days after injury, which is consistent with our data. In determining the expression of distinct microglia-related genes in vivo, ref. [23] showed that the expression of *C1qB*, *Galectin-3* and *p22^phox^* genes peaked on the 7 after the rat SCI with maximum values in the groups with mild and severe SCI. The products of the above genes are involved in pro-inflammatory signaling cascades [24]. However, there are no previous studies aimed at evaluating changes in the modulation of the microglia phenotype in the delayed periods after SCI. Ref. [24] only quantified the population of microglia/macrophages up to 180 days after rat SCI without establishing the secretory profile of these cells. At the same time, it was shown that the number of microglia/macrophages reaches a peak by day 60 and decreases linearly until day 180.

It is worth noting that the levels of pro-inflammatory cytokines such as IL-1α, IL-6, GRO/KC, RANTES, MIP-1α, MIP-3α and MCP-1 remained high in both the severe and mild SCI groups during the chronic period but were particularly high in the SCI 4 group. An increase in the above-mentioned cytokines was previously shown to be directly related to the development of neuroinflammatory processes, which can explain their sharp increase in the SCI 4 group [25,26,27,28]. However, for a number of chemokines, including RANTES, it was shown that a neuroprotective effect can be exerted precisely by their moderate increase, and neurotoxic properties are expressed only with a considerable rise in the expression level [29,30], which is consistent with our results. Thus, some cytokines have a dual role, positive in mild SCI and negative in severe SCI. We suggest that a similar trend may be true for other pro-inflammatory cytokines, whose expression remained elevated in both severe and mild SCI groups during the chronic period.

We observed the following pattern in the proliferative and phagocytic activity of microglia: the more severe the SCI, the lower the proliferation of microglia. This trend persisted when modeling SCI in vitro at 3, 14 and 60 dpi, which corresponded to the time periods when we observed the greatest changes in the phenotype of microglia depending on the degree of damage. It should be noted that the maximum proliferative and phagocytic activity of microglia were found in the model of the acute period (3 dpi) of mild SCI. It was previously shown that activated microglia can increase proliferation and phagocytic activity after SCI in rat and mice models [31,32], and up to 3 dpi is the main phagocytic cell type in the CNS [30]. There are data showing that by day 14 after mice SCI, peripheral macrophages become the main type of phagocytic cells [33]. However, there are no studies of the delayed periods after SCI, and in this regard, taking into account our results, it can be assumed that by 60 dpi, microglia again take on the main role. We would like to separately note that the decrease in the phagocytic activity of microglia that we found after 24 h of incubation with SCE may be associated with the physiological mechanisms of the gradual extinction of this process due to the saturation of these cells with the phagocytosed material contained in SCE, which was previously shown [34].

## 4. Materials and Methods

### 4.1. Microglia Isolation and Cultivation

Microglia were isolated from the cerebral cortex of 3-day-old Wistar rat pups. To replicate each experiment, one set of newborn rat pups, consisting of 7 to 10 pups, including both females and males, was used. Approximately 2 million microglia cells were isolated from each pup. After isoflurane anesthesia, intravital perfusion with DMEM/F12 medium was performed. The brain was extracted by cephalotomy and placed in 4 °C DMEM medium (BioloT, Saint-Petersburg, Russia) without serum. The cerebellum, olfactory bulbs and optic tracts were then removed, and the cerebral hemispheres were separated, followed by separation of the cortex from the underlying brain structures. Only the cerebral cortex was used for further work. The pia mater was removed, and the cortex matter was mechanically homogenized to fragments of 1 mm^3^.

The obtained homogenate was incubated with a solution of enzymes (papain 2 mg/mL, DNase 50 U/mL, BioloT) in DPBS for 15 min at 37 °C with shaking at 180 rpm. It was then mechanically homogenized by pipetting and incubated for 15 min under the same conditions and, again, homogenized by pipetting. The homogenate was passed through a 40 μm cell strainer. The resulting cerebral cortex homogenate was collected twice with DPBS from the enzyme solution by centrifugation for 10 min at 700× *g*. The resulting mixed cell suspension was centrifuged in an OptiPrep density gradient for 40 min at 300× *g* without sudden braking. The layers were cell suspensions in 19% OptiPrep, 9% OptiPrep, 7% OptiPrep and DPBS. A ring between 9% and 19% OptiPrep was carefully collected with a spinal needle and washed in a 10-fold volume of DPBS for 10 min at 700× *g*. After obtaining a microglia fraction, they were washed and cultured in DMEM/F12 medium supplemented with 10% fetal bovine serum (PAA Laboratories GE, Cölbe, Germany), 200 mM L-glutamine (PanEco, Moscow, Russia) and 1% Penicillin–Streptomycin (BioloT).

### 4.2. Immunocytochemistry

We also analyzed the phenotype of microglia at day 4–5 after isolation. Previously, light microscopy images were taken before staining with antibodies using an AxioObserver Z1 (Carl Zeiss, Jena, Germany). For immunocytochemistry (ICC), the cells were fixed in 4% buffered formalin, washed with 0.1% Triton X-100 in PBS and stained with GFAP (ab16997, Abcam, Cambridge, UK), Iba1 (ab5076, Abcam), TNF-α (ab199013, Abcam) and TGF-β (ab92486, Abcam) antibodies at a working dilution of 1:100 for 1.5 h at RT, then washed and stained secondary antibodies. For staining the appropriate Iba1, Alexa Fluor 488 conjugated anti-goat secondary antibodies ab150129 (Abcam), and for staining the appropriate GFAP, TNF-α and TGF-β, Alexa Fluor 647 conjugated anti-rabbit secondary antibodies A-31573 (Invitrogen, Waltham, MA, USA) were used. After incubation for 30 min at RT, the cells were counterstained with 40,6-Diamidino-2-phenylindole (DAPI) (10 mg/mL in PBS, Sigma, St. Louis, MI, USA) to visualize the nuclei. The results were analyzed using an LSM 780 Confocal Microscope (Carl Zeiss, Jena, Germany).

### 4.3. Spinal Cord Injury

All animal procedures were approved by the Kazan Federal University Animal Care and Use Committee (Permit Number: 2, dated 5 May 2015). Adult female Wistar rats weighing 250–300 g each were ordered from Pushchino Laboratory, Russia. Isoflurane anesthesia and intramuscular injection of Zoletil (20 mg/kg, Virbac Sante Animale, Carros Cedex, France) as the analgesia was administered immediately prior to surgery. The Th8 vertebra was removed by laminectomy next to skin incision. The impact rod of an impactor was placed above Th8 and dropped to induce SCI of varying severity: light (1.5 m/s, *n* = 20), moderate (2.5 m/s, *n* = 20) and severe (4 m/s, *n* = 20) (Impact One ™ Stereotaxic Impactor, Leica). Intact animals (*n* = 15) were used as control group. Animals exhibiting symptoms of systemic inflammation and without any signs of spinal cord injury were excluded from the experiment. The reported number of animals refers to the total number from which SCE were obtained. All post-surgical procedures were achieved according to the previously described protocol [35].

### 4.4. Spinal Cord Extracts (SCE) Isolation

After 3, 7, 14 and 60 dpi, the experimental animals were anesthetized by overdosed isoflurane anesthesia and intramuscular injection of Zoletil (20 mg/kg), followed by obtaining a 5 mm fragment of the spinal cord from the epicenter of the injury. Further manipulations were performed according to the previously described protocol [36]. The tissue fragments were homogenized with a Mini-BeadBeater-16 (BioSpec, Bartlesville, OK, USA) mill and zirconium beads (100 mg) in DPBS buffer for 40 s. The obtained SCE were centrifuged at 15,000× *g* for 15 min. Then, pipetted supernatants were filtered through a 0.22 μm PES filter to prevent contamination and prepared samples used as SCE for further studies. The PES filters were used as they are applicable for filtering protein solutions and culture media because of low extractables and low affinity for proteins. All tissue manipulation was carried out at 4 °C, and SCE samples were stored at −80 °C. Protein concentrations in the SCE samples were determined using the Pierce BCA kit Protein Assay Kit according to the manufacturer’s protocols (Thermo Scientific, Waltham, MA, USA).

### 4.5. Modeling of Spinal Cord Injury In Vitro

For modeling SCI in vitro, native microglia were cultured for 4–5 days until attaching to the culture plastic. All experiments were performed in five replications. The SCE of animals with varying severity of SCI obtained at 3, 7, 14 or 60 dpi was separately added to a culture of 4–5-day old microglia at a concentration of 500 µg/mL. Microglia cultured with the addition of intact spinal cord and on a normal medium (Medium,) were used as controls. The scheme of the experiment is shown in Figure 2.

### 4.6. Flow Cytometry

At 72 h after the addition of SCE samples of SCI 1.5, SCI 2.5 and SCI 4 obtained at 3, 7, 14 and 60 dpi, microglia samples were taken for flow cytometry (FC). Cells cultured on medium without the addition of SCE and with the addition of SCE from ISC were taken as controls. For FC, microglia cultures were trypsinized and incubated with antibodies to CD86 (BL305426, BioLegend, San Diego, CA, USA), CD45 (1,611,525, Sony, Tokyo, Japan) and CD206 (2,205,595, Sony) at a working dilution of 1 μL per 200,000 cells for 30 min at 4 °C. The stained cells were analyzed using a FACS Aria III (BD Biosciences, East Rutherford, NJ, USA) flow cytometer. The in vitro data were obtained from five independent experiments.

### 4.7. RNA Isolation and Real-Time PCR Analysis

The total RNA was isolated from microglia incubated with SCE for 72 h using the Yellow Solve Kit (Silex, Stockholm, Sweden) according to the manufacturer’s instructions. The RNA quantity was measured using NanoDrop (Thermo Scientific). Then, 100 ng of total RNA was reverse transcribed using 100 U of RevertAid reverse transcriptase (Thermo Scientific), 100 pmol of random hexamer primers and 5 U of RNAse inhibitor according to the manufacturer’s protocol (25 °C—10 min, 42 °C—60 min, termination of transcription 70 °C—10 min). A quantitative analysis of the mRNA of *Iba1*, *Il-1β*, *TGF-β*, *Il-6*, *TNF-α*, *CD209* and *CCL22* genes was performed using a CFX 96 Real-Time PCR System (Bio-Rad). The amplification conditions were the following: 95 °C—3 min, 39 cycles: 95 °C—10 s, 55 °C—30 s. Each reaction was performed in triplicate in a total volume of 10 μL and contained 100 ng of diluted cDNA, 2.5× Reaction mixture B (Syntol, Troy, Michigan), 200 nM of each primer and 100 nM SybrGreen (Eurogen, Singapore) (Appendix A). The mRNA expression was normalized, relatively, to 18S rRNA. The plasmid DNA with appropriate inserts was used to generate standard curves for quantification. To evaluate the copy number of the plasmid DNA insert, we used the DNA copy number calculator1. The Cq range for 18S was 13–14 cycles, at the same time as the test genes ranged from 25–31. The in vitro data were obtained from five independent experiments.

### 4.8. Cytokine Assay

To characterize the cytokine profile of the supernatants of the cultured microglia, cells were incubated with SCE for 72 h, and multiplex analysis using xMAP Luminex technology using the Bio-Plex Pro Rat Cytokine 23-Plex Immunoassay #12005641 (Bio-Rad, Hercules, CA, USA) was used. This kit allows us to set the level of 23 cytokines and chemokines: G-CSF, GM-CSF, GRO/KC, IFN-γ, Il-1α, Il-1β, Il-2, Il-4, Il-5, Il-6, Il-7, Il-10, Il-12 (p70), Il-13, Il-17A, Il-18, M-CSF, MCP-1, MIP-1α, MIP-3α, RANTES, TNF-α and VEGF. The in vitro data were obtained from five independent experiments.

### 4.9. Microglia Proliferation Assay

The proliferative activity of microglia was evaluated in the model of SCI in vitro with varying severity at different periods. First, 50,000 cells per well and SCE at a concentration of 500 μg/mL in E-plates were placed into the xCELLigence Real-Time Cell Analysis system (LaRoche, Basel, Switzerland), which captures changes in cell behavior in real time using microelectrodes. The above-mentioned analysis was carried out within 96 h. The in vitro data were obtained from five independent experiments.

### 4.10. Phagocytic Activity

The phagocytic activity of the microglia was assessed under the conditions of modeling different periods of varying severity in SCI in vitro. It was carried out after 1, 24 and 72 h of cultivation with SCE. The object of phagocytosis was live baker’s yeast *S. cerevisiae*, previously labeled with trypan blue dye. The baker’s yeast was previously inactivated at 100 °C for 30 min in DPBS, then, stained with trypan blue and washed thoroughly to remove excess dye by repeated centrifugation. Then, 5,000,000 pre-stained yeast cells were added to the microglia culture after cultivation with SCE. After incubation, the wells were washed thoroughly with DPBS to remove excess yeast particles. Trypan blue was eluted using lysis buffer (70% ethanol, 0.1% glacial acetic acid). The optical density of the eluents was evaluated on an Infinite Pro 2000 (Tecan, Männedorf, Switzerland) with a wave length of 540 nm. The in vitro data were obtained from five independent experiments.

### 4.11. Statistical Analysis

Data were analyzed using Origin 7.0 SR0 Software (OriginLab, Northampton, MA, USA). The results created were expressed as a mean ± standard deviation (SD). A one-way analysis of variance (ANOVA) with Tukey’s test was used for various comparisons between all experimental and control groups. The Mann–Whitney test was applied to analyze the results of the PCR analysis. All analyses were against the treatment group. The mean differences between experimental groups were measured as significant if *p* < 0.05 was reached.

## 5. Conclusions

In our study, we evaluated the behavior of microglia under the conditions of modeling mild, moderate and severe SCI in vitro for various post-traumatic periods (acute, subacute and chronic) for the first time. We characterized changes in the phenotypic characteristics, gene expression profile, various cytokine expressions and the proliferative and phagocytic activity of these cells, taking into account the existing changes in their microenvironment. Our data indicate active changes in the behavior of microglia depending on the degree of damage and the presence of prolonged activation of the investigated cells in the chronic period after severe SCI. We clarify that we cannot talk about a specific reactive state since we found that microglia make a significant contribution to both anti- and pro-inflammatory patterns in severe SCI. In this regard, the question remains whether the above-mentioned activity is associated with an attempt by microglia to normalize the lingering consequences of severe injury or whether this is one of the direct limiting factors of regeneration, which must be therapeutically addressed. Unfortunately, we cannot yet answer this question, but we hope that subsequent studies will lead to its disclosure.

## Figures and Tables

**Figure 1 ijms-24-08294-f001:**
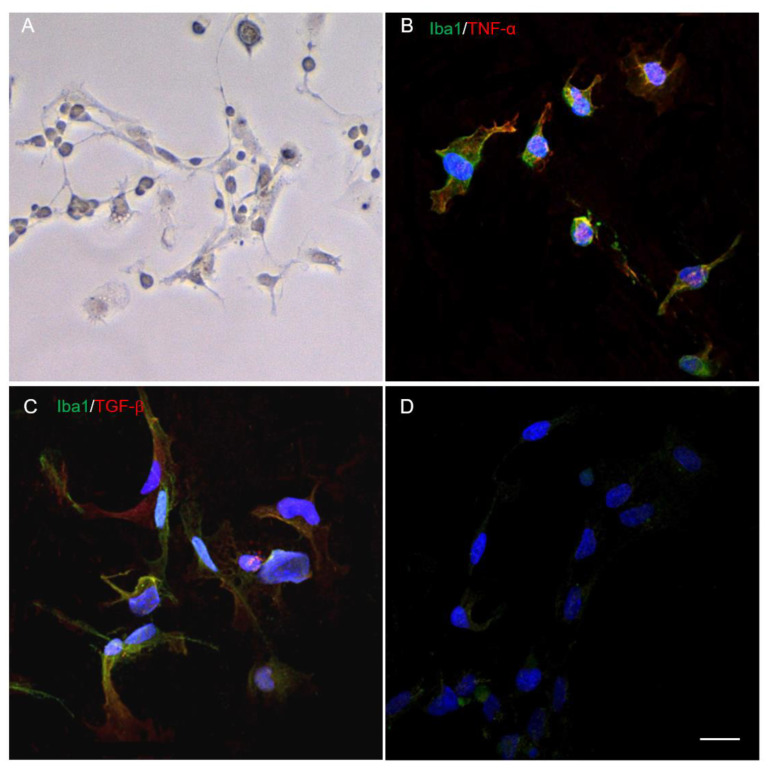
Visualization of microglia using light microscopy (**A**). Immunocytochemistry demonstrates the expression of Iba1 (green, (**B**,**C**)), TNF-α (red, (**B**)), TGF-β (red, (**C**)) and negative control (**D**). Nuclei are stained with DAPI (blue). Scale bar: 20 µm.

**Figure 2 ijms-24-08294-f002:**
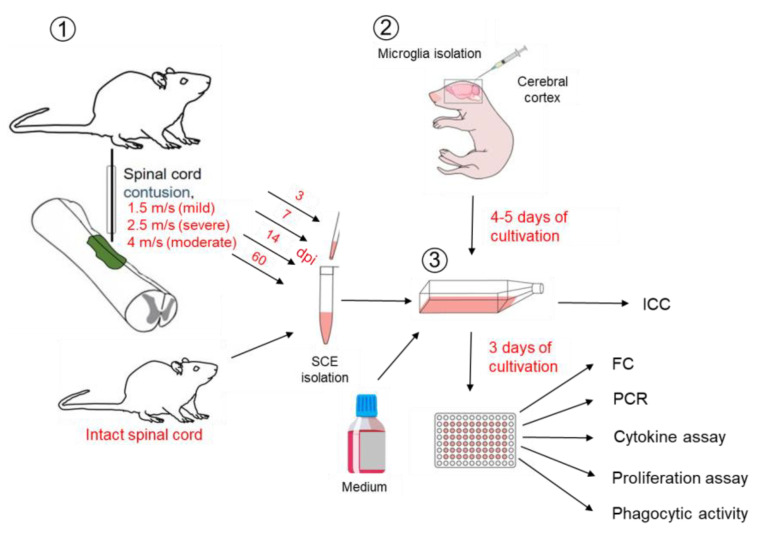
Experiment scheme. Spinal cord extracts (SCE) of animals with SCI of varying severity obtained at 3, 7, 14 or 60 dpi (1). Microglia isolated from the cerebral cortex of 3–4 days old rat pups (2). After 4–5 days of cultivation, part of the cells were analyzed by immunocytochemistry (ICC). Further SCE separately added to a culture of 4–5-day old microglia at a concentration of 500 µg/mL (3). Microglia cultured with the addition of intact spinal cord (ISC) and on a normal medium (Medium) were used as controls. At 72 h, microglia samples were taken for FC, PCR, cytokine assay (supernatants), proliferation assay, phagocytic activity.

**Figure 3 ijms-24-08294-f003:**
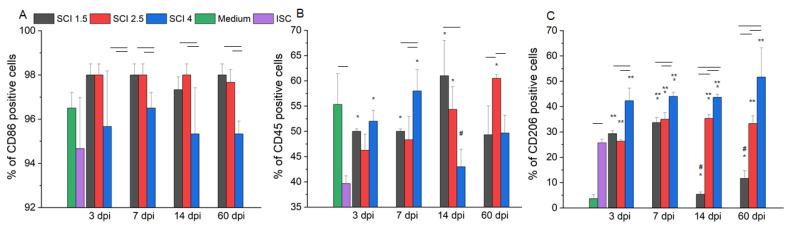
Flow cytometry of microglia on the model of SCI in vitro. Percent of microglia expressing CD86 (**A**), CD45 (**B**) and CD206 (**C**) after cultivation with medium (green column) and SCE obtained from intact spinal cord (purple column) and at 3, 7, 14 and 60 days after mild SCI 1.5 (black column), moderate SCI 2.5 (red column) or severe SCI 4 (blue column). SCE was added to microglia on 4–5 days after isolation and after the 72 h analysis was performed. Horizontal lines illustrate the presence of a significant difference (*p* < 0.05) between the groups. * *p* < 0.05 as compared with ISC group. ** *p* < 0.05—as compared with Medium group. # *p* < 0.05 as compared with relevant indicators at 3 and 7 dpi.

**Figure 4 ijms-24-08294-f004:**
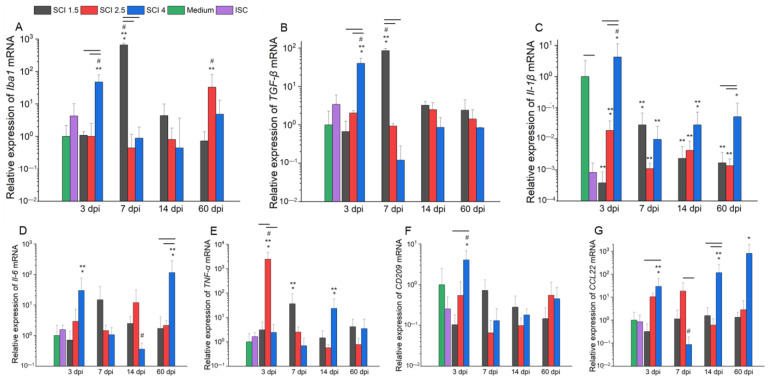
Analysis of mRNA expression in microglia on the model of SCI in vitro. *Iba1* (**A**), *TGF-*β (**B**), *Il-1*β (**C**), *Il-6* (**D**), *TNF-*α (**E**), *CD209* (**F**) and *CCL22* (**G**) mRNA expression of microglia cultivated with medium (green column) and SCE obtained from intact spinal cord (purple column) and at 3, 7, 14 and 60 days after mild SCI 1.5 (black column), moderate SCI 2.5 (red column), or severe SCI 4 (blue column), calculated in relation to medium control, which was considered as 1. SCE was added to microglia on 4–5 days after isolation and after the 72 h analysis was performed. Horizontal line means significant difference (*p* < 0.05) between groups. * *p* < 0.05 as compared with the ISC group. ** *p* < 0.05 as compared with Medium group. # *p* < 0.05 as compared with relevant indicators in all investigated dpi.

**Figure 5 ijms-24-08294-f005:**
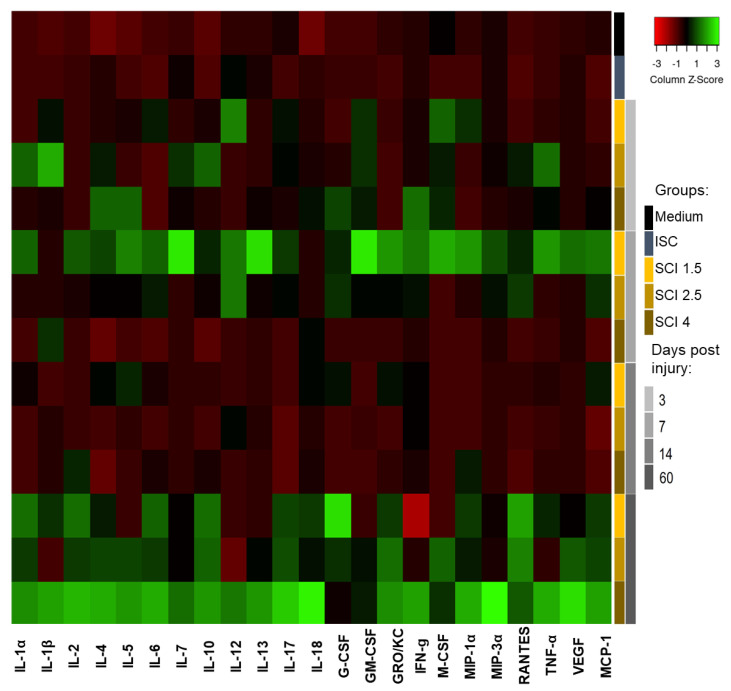
Heat map analysis of the microglia supernatants cytokine profile after cultivation with Medium and SCE obtained from intact (ISC group) and injured spinal cord (SCI 1.5, SCI 2.5 and SCI 4 groups) at different dpi.

**Figure 6 ijms-24-08294-f006:**
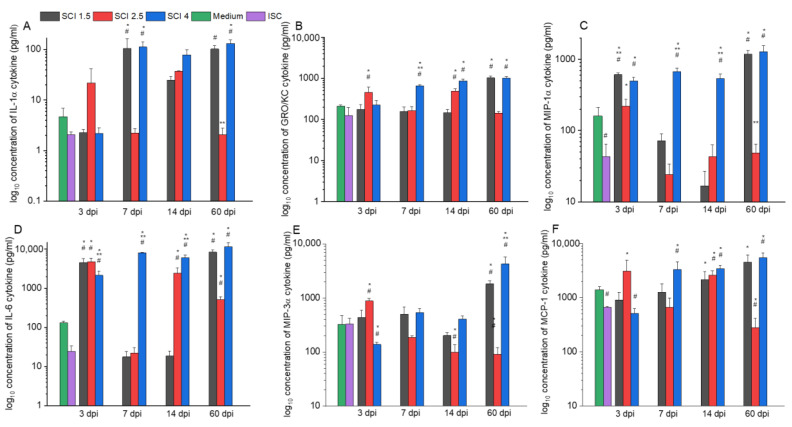
Cytokine profile of microglia supernatants on the model of SCI in vitro. Concentration of IL-1α (**A**), GRO/KC (**B**), MIP-1α (**C**), IL-6 (**D**), MIP-3α (**E**) and MCP-1 (**F**) cytokines in supernatants of microglia cultivated with medium (green column) and SCE obtained from intact spinal cord (purple column) and at 3, 7, 14 and 60 days after mild SCI 1.5 (black column), moderate SCI 2.5 (red column) or severe SCI 4 (blue column). SCE was added to microglia on 4–5 days after isolation and after the 72 h analysis was performed. ** p* < 0.05 as compared with the ISC group; *# p* < 0.05 as compared with the Medium group; *** p* < 0.05 as compared with other groups at the same dpi.

**Figure 7 ijms-24-08294-f007:**
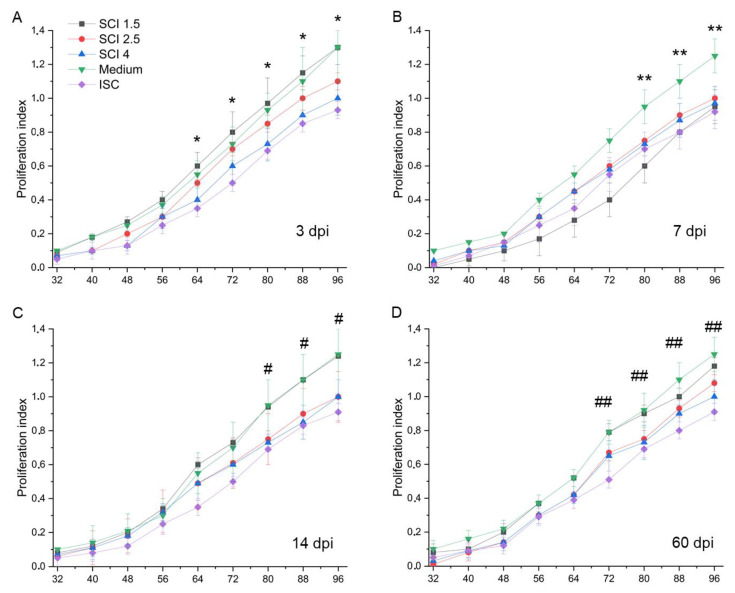
Analysis of microglia proliferation on the model of SCI in vitro. Proliferative activity of microglia cultivated with medium (green line) and SCE obtained from intact spinal cord (purple line) and at 3 (**A**), 7 (**B**), 14 (**C**) and 60 (**D**) days after mild SCI 1.5 (black line), moderate SCI 2.5 (red line) or severe SCI 4 (blue line). SCE was added to microglia 4–5 days after isolation and incubation for 96 h. X axis—time in hours, Y axis—proliferation index of microglia. The lack of data up to 32 h is associated with the absence of changes in microglia proliferation from 0 to 32 h of monitoring. * *p* < 0.05 for the SCI 1.5 group compared to the SCI 4 and ISC groups. ** *p* < 0.05 for the Medium group compared to other investigated groups. # *p* < 0.05 for the Medium group compared to the SCI 2.5, SCI 4 and ISC groups. ## *p* < 0.05 for Medium group compared to ISC group.

**Figure 8 ijms-24-08294-f008:**
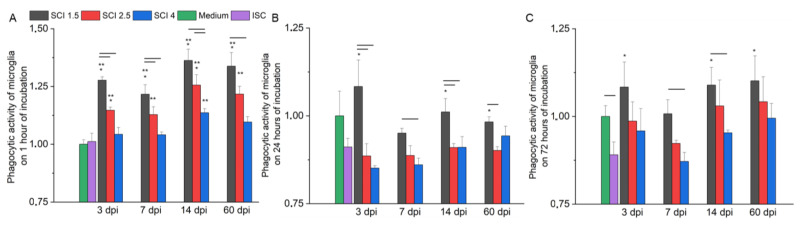
Analysis of microglia phagocytic activity on the model of SCI in vitro. The phagocytic activity of microglia cultivated with medium (green line) and SCE obtained from intact spinal cord (purple column) and at 3, 7, 14 and 60 days after mild SCI 1.5 (black column), moderate SCI 2.5 (red column) or severe SCI 4 (blue column), calculated in relation to the Medium control, which was considered as 1. SCE was added to microglia on 4–5 days after isolation, and phagocytic activity was analyzed at 1 (**A**), 24 (**B**) and 72 (**C**) hours of incubation. Horizontal line means significant difference (*p* < 0.05) between groups. * *p* < 0.05 as compared with ISC group. ** *p* < 0.05 as compared with Medium group.

## Data Availability

Data available upon request from the authors.

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
