# Peer review of "Severity- and Time-Dependent Activation of Microglia in Spinal Cord Injury"

_ijms, 2023, doi:10.3390/ijms24098294_

Round 1

Reviewer 1 Report

The authors have studied the influence on rat microglia cultures exposed to extracts from a spinal cord contusion site of adult rats at different postinjury periods, and following spinal cord injuries induced by different intensity of physical impact. A wide range of assessment methods have been used to determine how the different experimental conditions affect microglia features of pro- versus anti-inflammatory. The authors demonstrate a temporal and severity related pattern of microglia reactivity, which is not simply falling into simplified states of M1 or M2 activation. The study is well designed and documented, and with adequate interpretation and discussion of the data.

Minor comments

1. Figure 1 is quite helpful, but should be placed in the beginning of Results. It is difficult to follow this section without this Figure, and there is no reason to place I far down in the manuscript.

2, The abbreviation ICC in Figure 1 is not explained, whereas ISC in the Figure legend is absent in the Figure. Perhaps just a confusion of letters.

Author Response

We would like to thank the reviewer for this review and pointing out these comments. According to the reviewer’s comment, we have made the appropriate changes.

1. We agree with your opinion that the drawing with the scheme of the experiment turned out to be too down in the Manuscript due to the transfer of the Materials and Methods chapter to the end of the article. The figure has been moved to the top of the text.

2. You are right, the abbreviation ICC should have been introduced for the clarity of the Figure. We added an explanation in the Figure legend. All revisions made to the manuscript are marked up using the “Track Changes” function and you can easily view it. We hope that our responses will satisfy the reviewer.

Reviewer 2 Report

The manuscript, submitted from a team of authors led by a well-established SCI researcher, is very interesting and a highly topical. The authors  investigated the pro- and anti-inflammatory involvement of  SCI,  depending on its severity and the post-traumatic periods (3,7,14 and 60 dpi). They provide valuable results which facilitates the understanding of the reactive states of microglial cells in the microenvironment of SCI. Overall, the study is well conducted with numerous results.

 I have only some minor comments:

Figs. 3-5 and 7 images  are not legible in the printed version. Please make correction. The abbreviation ISC is explained in subsection Material and Methods, but appears much earlier in the text. In Introduction and Discussion- Try to clarify which animal species were used when quoting a specific findings from an in vivo study.

Line 42- Please change  “cytokines tumor” to “cytokines, such as  tumor”

Author Response

We would like to thank the reviewer for this review and pointing out constructive comments. We have resized the captions on the figures to be larger in the figures 3-5 and 7. In addition, Figure 5 has been split into two separate images for clear visualization. According to your comments, abbreviations were introduced in the right place. Due to the format of the journal, Materials and Methods section ended up at the end of the text, and therefore all abbreviations ended up at the end of the Manuscript. We fixed this error.

In Introduction and Discussion we clarified which animal species were used when quoting a specific findings from an in vivo studies. Made changes to line 42: changed “cytokines tumor” to “cytokines, such as  tumor”. All revisions made to the manuscript are marked up using the “Track Changes” function and you can easily view it. We hope that our responses will satisfy the reviewer.